# Waste Utilization: Insulation Panel from Recycled Polyurethane Particles and Wheat Husks

**DOI:** 10.3390/ma12193075

**Published:** 2019-09-20

**Authors:** Štěpán Hýsek, Pavel Neuberger, Adam Sikora, Ondřej Schönfelder, Gianluca Ditommaso

**Affiliations:** 1Department of Wood Processing and Biomaterials, Faculty of Forestry and Wood Sciences, Czech University of Life Sciences Prague, Kamýcká 129, 165 00 Prague, Czech Republic; sikoraa@fld.czu.cz (A.S.); schonfelder.ondrej@seznam.cz (O.S.); ditommaso@fld.czu.cz (G.D.); 2Department of Mechanical Engineering, Faculty of Engineering, Czech University of Life Sciences Prague, Kamýcká 129, 165 00 Prague, Czech Republic; neuberger@tf.czu.cz

**Keywords:** heat insulation, polyurethane, husk, recycling, waste

## Abstract

This study provides a solution for the utilization of two waste materials, namely the residues of soft polyurethane foam from the production of mattresses and winter wheat husks. Thermal insulation panels with a nominal density of 50–150 kg/m^3^, bonded one-component moisture curing polyurethane adhesive, were developed, and the effect of the ratio between recycled polyurethane foam and winter wheat husk on internal bond strength, compressive stress at 10% strain, water uptake, coefficient of thermal conductivity, and volumetric heat capacity was observed. The developed composite materials make use of the very good thermal insulation properties of the two input waste materials, and the coefficient of thermal conductivity of the resulting boards achieves excellent values, namely 0.0418–0.0574 W/(m.K). The developed boards can be used as thermal insulation in the structures of environmentally friendly buildings.

## 1. Introduction

Nowadays, energetic efficiency, thermal insulation, and eco-compatibility are fundamental properties of modern buildings. Energy efficiency is strongly connected to the development of a building and the thermal insulation is one of the keys to increasing it. Scientific research is not focusing solely on the low value of thermal conductivity, but also on the development and use of natural materials. A new trend in this field is the use of recycled materials, for their convenience on the market and especially for their potential to be used for insulation panels in new buildings and reconstructions. The largest part of a building’s energy consumption can be attributed to the operating phase, influenced by several factors, such as window and door thermal insulation [1,2] and opaque wall thermal performance [3]. When thinking about insulation materials, it can be assumed that inside the wall structure these are the layers that contribute most to the overall thermal behavior of the walls during the warmest and coldest months. This is directly linked to the external conditions with its specific thermo-physical properties [4]. The increased investments in near-zero buildings is also promoting the use of passive solutions for the envelope, resulting in increased insulation thicknesses of walls [5].

In terms of consumption of resources and waste generation, the construction sector is among those of the economy with the highest impact on the environment. In a sustainable construction approach, choosing the right insulating material must include a correct analysis of the entire life-cycle of the product. Nowadays, the use of natural fillers for the reinforcement of composites has received increasing interests from academics and industry. Many kinds of natural resources have been analyzed for industrial utilization, such as: flax, hemp, wood, wheat, barley, and oats [6,7,8]. These studies are now evolving quickly as reasonable alternatives to synthetic materials for different applications, for example in building materials or automotive components [9,10,11,12].

Husk material (husk from rice or wheat) can be suitable for the fabrication of different types of panels for building applications [8]. Its acoustic and thermal behavior can be compared to many other panels with a different filler–matrix. If we think about husk as filler, we can assume that it has many advantages over mineral fillers; because it is a non-abrasive material, it requires less energy for processing, and it can reduce the density of the final furnished products. The physical and mechanical properties of a natural filler are strongly dependent on the matrix type, content, and properties of the fillers used for reinforcement. One of the most important reasons to use a natural fiber or filler-reinforced composite materials is susceptibility to moisture absorption and the resultant effect of this on the physical, mechanical, and thermal properties [13]. However, the effect of moisture absorption leads to the degradation of the filler–matrix interface region, creating poor stress transfer efficiencies and ending in a reduction of the mechanical properties of the panel [8,14].

In this paper, husks are combined with polyurethane (PUR) recycled particles, and we hypothesize that through a combination of husks and PUR particles, the disadvantages of wheat husks can be eliminated. This study is focused on the production of heat insulation panels from recycled materials and analyses from different points of view to better understand the thermal and moisture behavior of this innovative composite material. The effect of the ratio between wheat husks and PUR particles on the observed properties is studied.

## 2. Materials and Methods

Recycled open cell flexible polyurethane (PUR) foam with a density of 24 kg/m^3^ and a bulk density of 11.3 kg/m^3^ and winter wheat husks were used to produce insulating boards. One-component moisture-curing polyurethane adhesive NEOPUR M 2238 R Agglu was used as an adhesive, and the weight of the adhesive ratio in all the boards was 20%. According to the technical data sheet, the density of used adhesive was 1.13 g/m^3^ and the viscosity at 25 °C was 2500 mPas. The mutual ratio of husks and PUR particles was variable and the individual variants are shown in Table 1. In Table 2 are presented actual production parameters. The plan dimensions of one board were 400 mm × 400 mm and the required thickness was 50 mm. The moisture content of each material was measured using the OHAUS MB 23 device (Ohaus Corporation, Parsippany, NJ, USA) in order to determine the correct amount of dry matter. The moisture content of the PUR particles was 2.3% and moisture content of the husks was 7%. The adhesive was applied using a laboratory adhesive applicator. The resulting mixture of PUR particles, husks, and adhesive was uniformly layered into the wooden mold covered by polyethylene film. The boards were pressed for one day under laboratory conditions. The target thickness parameter of 50 mm was affixed to all of the produced boards. Due to the higher bulk density of the husks than the PUR particles, boards with a higher proportion of husks achieved a higher average board density.

### 2.1. Measuring Thermal Insulation Properties

Thermal insulation properties were measured at 23 °C and relative humidity (RH) 36% using device Isomet 2104 (Applied Precision, Ltd., Bratislava, Slovakia). A total of 4 boards were measured from each variant and 3 measurements were carried out on each board. The measurements were carried out using a needle probe (measuring range 0.015–2 W/(m.K)), and the probe was used in accordance with the manufacturer’s recommendations. The monitored characteristics were the coefficient of thermal conductivity and volumetric heat capacity.

### 2.2. Determination of Short-Term Water Absorption at Partial Submersion

In order to test the short-term water uptake at partial submersion according to [15], ten samples were prepared from each insulation board variant, each having plan dimensions of 100 mm × 100 mm, the thickness being the same as the thickness of the produced boards. After weighing the samples m_0_, they were placed in a test tank on a grid that allowed water to pass through. The specimens were encumbered in such a way that they did not float in the water and water was added so that the bodies were submerged 10 mm below the surface. The level was checked regularly, and water was topped up as necessary. After 24 h, the specimens were removed from the tank and placed on a grid that was at an inclination angle of 35°–40°. Here the samples were allowed to drain for 10 min. The weight m_24_ was then measured for the samples. Water uptake per square meter of board (kg/m^2^) and water uptake per dry mass of board (%) were determined according to the following formulas:(1)$$W_{sq}=\frac{m_{24}-m_{0}}{A_{p}}$$
(2)$$W_{m}=\frac{m_{24}-m_{0}}{m_{0}}\cdot 100\%$$

$W_{sq}$ is the water uptake per square meter of board (kg/m^2^)

$W_{m}$ is the water uptake per dry mass of board (%)

$m_{24}$ is the weight of the test specimen after 24 h of partial submersion (kg)

$m_{0}$ is the initial weight of the test specimen (kg)

$A_{p}$ is the lower surface area of the test specimen (m^2^)

### 2.3. Determination of Tensile Strength Perpendicular to the Level of the Board

From each insulation board variant, ten test specimens with dimensions of 50 mm × 50 mm were designed to test the board internal bond strength. The tensile strength perpendicular to the plane of the boards was determined according to [16] using universal tensile testing machine TIRA test 2850 (TIRA GmbH, Schalkau, Germany).

### 2.4. Determination of Compressive Strength Perpendicular to the Level of the Board

Ten specimens from each 100 mm × 100 mm board variant were tested for compressive strength at 10% compression from the original height. The samples loaded into the test machine were initially encumbered with 2 N preload and the test was then started. The determination of compressive strength perpendicular to the level of the board was carried out according to [17] using universal tensile testing machine TIRA test 2850. Compressive stress at 10% strain was calculated according to the following formula:(3)$$\sigma_{10}=\frac{F_{10}}{A_{0}}$$

$\sigma_{10}$ is compressive stress at 10 % strain (MPa)

$F_{10}$ is force at 10% deformation (N)

$A_{0}$ is the initial cross-sectional area of the test specimen (mm^2^)

### 2.5. Statistical Analysis

Descriptive statistics and analysis of variance were used to characterize the data. The Tukey post-hoc test was used to determine if any of the differences between the pairwise means were statistically significant. A significance level of α = 0.05 was selected. The impact of factor PUR: husks ratio on the observed characteristics was shown graphically, and the vertical columns represent the 95 percent confidence intervals.

## 3. Results and Discussion

Figure 1 shows a cross-section of the manufactured panels showing a different proportion of PUR particles and husks. Table 3 shows the measured characteristics of the manufactured insulation boards.

Figure 2 shows the results of the thermal conductivity coefficient (λ). The results show that the overall thermal conductivity coefficient increased with increasing husks, which also corresponds to the density of the individual materials. The best variant was therefore the type of boards without added husks (λ = 0.0418 W/(m.K)). Boards with 20% husks achieved higher values on average by 3.83% (λ = 0.0434 W/(m.K)) compared to the variant without added husks. For boards with 40% husks, higher values were measured relative to the reference board by 8.13% (λ = 0.0452 W/(m.K)). From a statistical point of view (Table 4), it can be stated that there is no statistically significant difference between variants without added husks and with 20% husks. The same is true for boards with 20% versus 40% husks. For materials with a higher proportion of husks than foam, the thermal conductivity coefficient increased significantly. For material with 60% husks, an increase in value relative to the reference material by 18.9% (λ = 0.0497 W/(m.K)) was recorded. For material made up of husks only, the measured values reached an average of 37.32% (λ = 0.0574 W/(m.K)) higher value relative to the reference material. The described trend is closely related to the density of the individual variants, in general the thermal conductivity coefficient decreases with the decrease of the foam density down to ca. 80 kg/m^3^ [18,19]. Increasing the proportion of husks in the composition of the material significantly increases the coefficient of thermal conductivity; nevertheless, this material is comparable to other commonly used materials such as cork boards, c-flute panels, etc. [20]. The measured thermal conductivity coefficients are comparable to commercial wood fiber heat insulation boards, for example, from Steico company.

A similar trend to that of the thermal conductivity can also be seen in volumetric heat capacity. This trend can be seen in Figure 3. The lowest volumetric heat capacity values were measured in materials without added husks, and with 20% and 40% husks added. There was no statistically significant difference between these materials (Table 5). However, it was found that as the volume of husks in the boards increases, the volumetric heat capacity also increases. For boards with 20% husks (*C*_p_ = 0.108 MJ/m^3^K), an increase in values was recorded relative to the boards without husks (*C*_p_ = 0.105 MJ/m^3^K) at an average of 2.86%. For boards with 40% husks (*C*_p_ = 0.113 MJ/m^3^K), relative to the boards without husks, an increase in values was recorded at an average of 7.62%. For boards with 60% (*C*_p_ = 0.154 MJ/m^3^K) and 80% husks (*C*_p_ = 0.237 MJ/m^3^K), the increase was statistically significant. For boards with 60% husks, relative to the boards without added husks, the increase was at an average of 46.67%, and for boards with 80% husks, the increase was at an average of 125.71%. The measured volumetric specific heat capacities of the board without added husks correlates to the results specified in the work of Incropera et al. [21]. Boards containing husks had generally higher values of volumetric heat capacity with an increasing proportion of husks in the board. However, boards containing 80% husks achieved much lower volumetric heat capacity values than boards from the husks specified in the work of Czajkowski et al. [22]. The reason for this difference may be a difference in the density of the boards (Table 3).

The results of internal bond strength are shown in Figure 4. The highest values were measured for boards without added husks (31.1 kPa) and with the increasing number of husks, internal bond strength also decreased significantly, but this drop was only true for boards with less than 40% husks. There was no statistically significant increase or decrease in values from this threshold (Table 6. Compared to the boards without husks, boards with 20% husks (21.5 kPa) decreased on average by 30.87%. Compared to the boards without husks, in boards with 40% husks (3.4 kPa), the decrease was at an average of 86.17%. Boards with 60% husks (2.3 kPa) showed a decrease at an average of 92.60% and boards with 80% husks (5.8 kPa) showed a decrease in values at an average of 81.35%. Due to the fact that the internal bond strength of boards with a higher proportion of husks and higher density was lower than for boards with significantly lower density, it can be stated that the interaction of the adhesive, foam, and husks significantly negatively affects the internal bond strength values. This trend is explained by the diametrically different adhesive properties of foam and husks. With an increasing proportion husks, the internal bond strength values stabilized at values corresponding to boards consisting of 80% husks. Similar results were achieved in the work [23]; however, wood fibers were used in this work. If we compare our result with high density binderless insulation boards made from coconut husks and bagasse in work [24], it can be seen that results in this work are very similar. It follows that it is possible to produce an insulation material only from husks without adhesive, but at the expense of higher board density.

The results of the compressive stress at 10% stress can be seen in Figure 5. From these results, there is an apparent trend of increasing stress at 10% strain with increasing husks content. The highest stress values were measured for boards with 80% husks (18.3 kPa). Boards without husks (2.9 kPa) achieved lower values relative to boards with 80% husks at an average of 84.15%. Using the statistical evaluation in Table 7, it can be seen that there was a statistically insignificant difference (for boards with 60% and 40% husks, there was a slight statistically significant difference) between boards without husks and boards with up to 60% husks. Boards with 20% husks (2.7 kPa) achieved lower values relative to the boards without added husks on average by only 7.41%. Boards with 40% husks (1.4 kPa) achieved lower values compared to boards without husks at an average of 51.72%. Adversely, boards with 60% husks (5.5 kPa) achieved higher average values compared to the boards without husks by 89.66%. These results are comparable to those specified in the work [25]. This work dealt with bio polyurethane foams. However, there were significant differences in board density between our materials and those presented in this work.

Figure 6 show the results of the short-term water uptake of the boards at partial submersion. Two approaches were used for evaluation. The first approach (Figure 6 left) according to standard [15] is the expression of water uptake relative to the area in kg/m^2^. These results show no statistically significant difference for all of the tested materials (Table 8). The water uptake values averaged values ranging from 4.32–3.75 kg/m^2^. The water uptake was also reduced with the number of husks added. The measured results are comparable, for example, to wood fiber-based materials with the addition of a mediator (low molecular weight material 4-hydroxybenzoic acid (HBA) with a chemical purity of 99%) specified in the work of Kirsch et al. [23]. Due to inconclusive evidence, we decided to express the water uptake as a proportion of the board dry matter, and the results can be seen in Figure 6 right. From a statistical point of view, it can be stated that there were no statistically significant differences after the 40% proportion of husks, but there was a statistically significant decrease in values (Table 9) for the transition between 40% and 60% husks. For boards without added husks, the average values were around 132.4%. For boards with 20% husks (131.2%) there was a decrease relative to the boards without added husks by 0.91%, and for boards with 40% husks (123.6%) there was a decrease of 6.65%. Boards with 60% husks (55.1%) achieved a more significant decrease relative to the boards without added husks by 58.38%. The biggest difference was measured for boards with 80% husks (46.2%). Mixing the open cell polyurethane foam with husks (components with lower content of opened cells) reduced the ratio of open cells to closed cells and thus generally reduced the water uptake [26,27].

The microscopic images in Figure 7a–c show the observed characters of rupture in the material after the tensile strength test perpendicular to the plane of the board. Both adhesion errors and cohesion errors were observed. Adhesion failure occurred when the husks and PUR particles came into contact (Figure 7a,c). Cohesive failure occurred in PUR particles (Figure 7b). The separation of the adhesive from the particle surface is an undesirable phenomenon in the composites—in our case it is caused by waxy substances on the surface of the husks and this adhesion error could be eliminated by a suitably chosen surface pre-treatment [8].

## 4. Conclusions

The developed waste material panels showed excellent thermal insulation properties. The thermal conductivity coefficient increased with increasing husks, and its maximum value of 0.0574 W/(m.K) was achieved by boards only from husks with a density of 153 kg/m^3^. However, with the increasing thermal conductivity coefficient, the heat capacity also increased, which can be seen as positive. The effect of the PUR/husks ratio on internal bond strength was shown to have a poor interaction between husks and PUR adhesive. In general, it can be stated that with the increase in the husk ratio, there was a significant decrease in internal bond strength values. The results of compressive stress at 10% strain correlated with density. Boards with increasing density thus achieved higher values of compressive stress at 10% strain. The developed boards exhibited high water uptake, which makes them suitable for use as thermal insulation in wall compositions, where they will be protected against moisture by structural protection.

## Figures and Tables

**Figure 1 materials-12-03075-f001:**
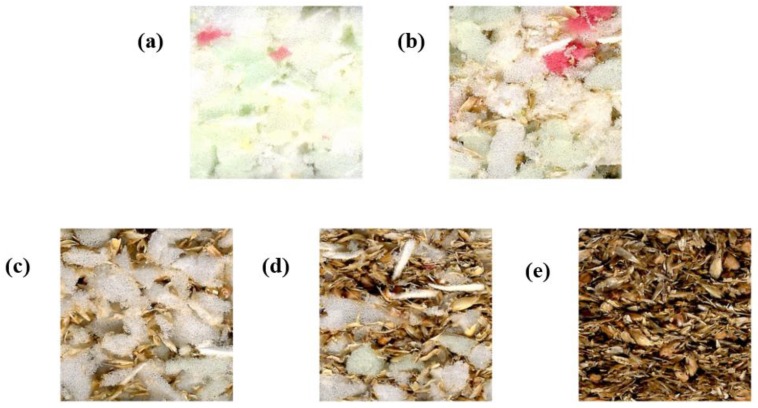
Cross section of the panels. Ratio foam: husks: adhesive: (**a**) 80:0:20; (**b**) 60:20:20; (**c**) 40:40:20; (**d**) 20:60:20; (**e**) 0:80:20.

**Figure 2 materials-12-03075-f002:**
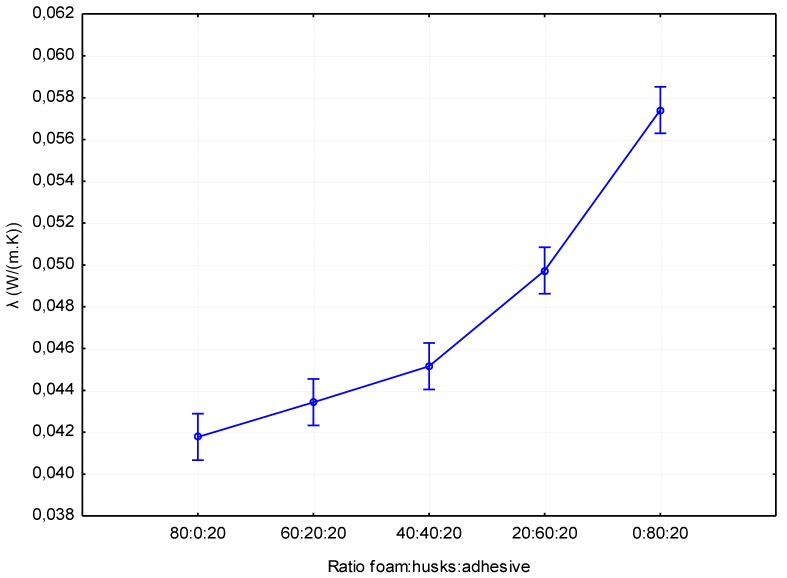
Influence of panel composition on the thermal conductivity coefficient.

**Figure 3 materials-12-03075-f003:**
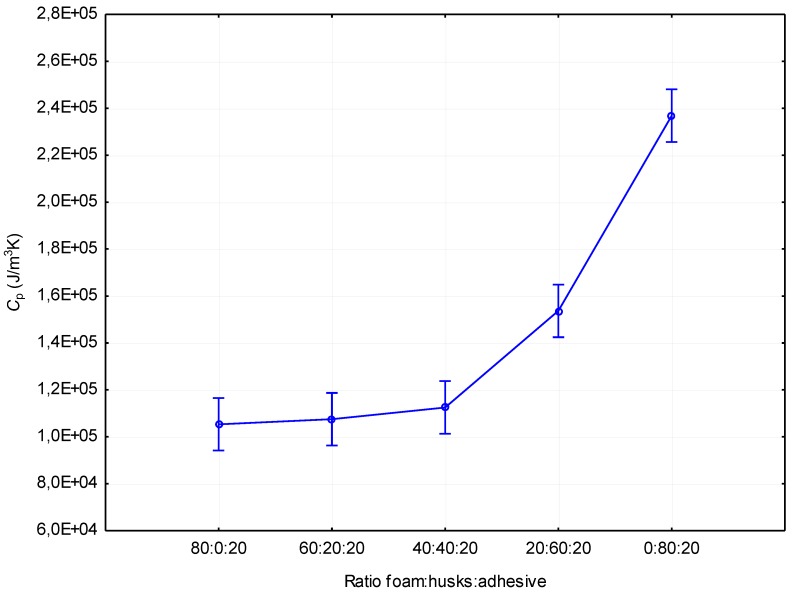
Impact of the composition of the panel on volumetric heat capacity.

**Figure 4 materials-12-03075-f004:**
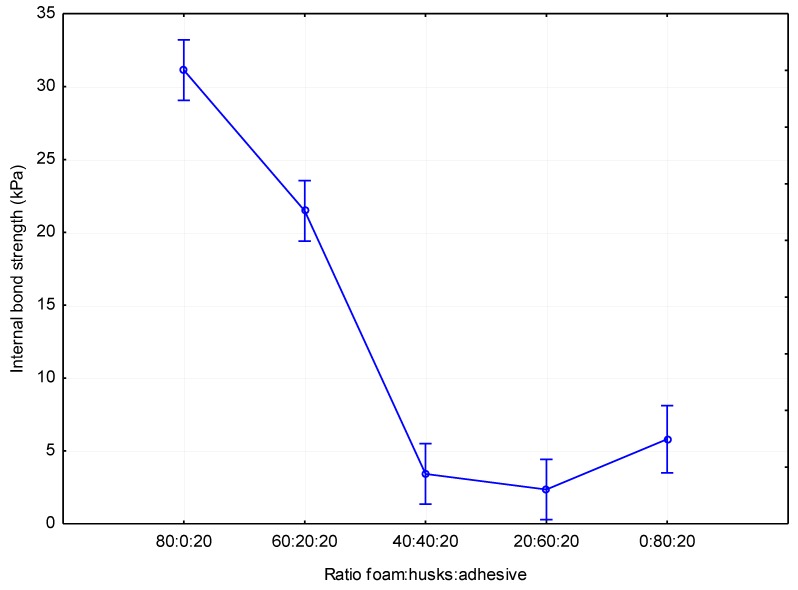
Impact of the composition of the panel on internal bond strength.

**Figure 5 materials-12-03075-f005:**
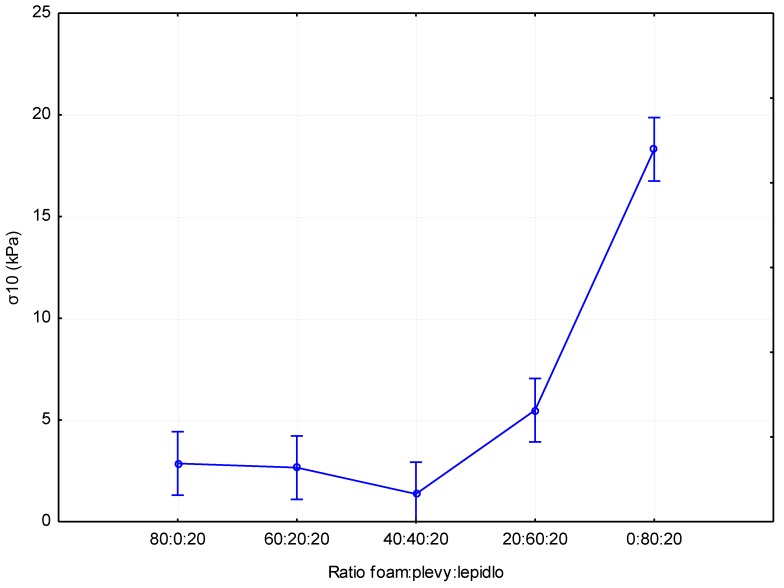
Impact of the composition of the panel on stress at 10% compression.

**Figure 6 materials-12-03075-f006:**
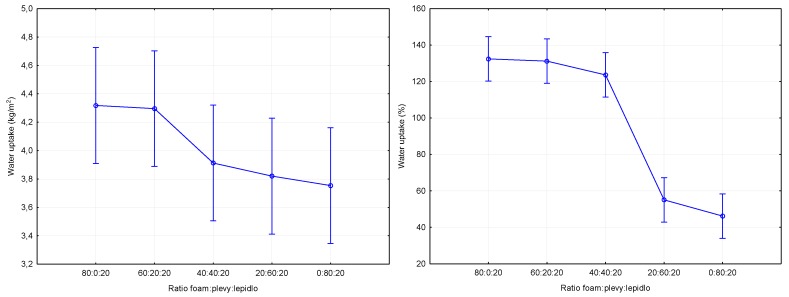
Impact of the composition of the panel on short-term absorption.

**Figure 7 materials-12-03075-f007:**
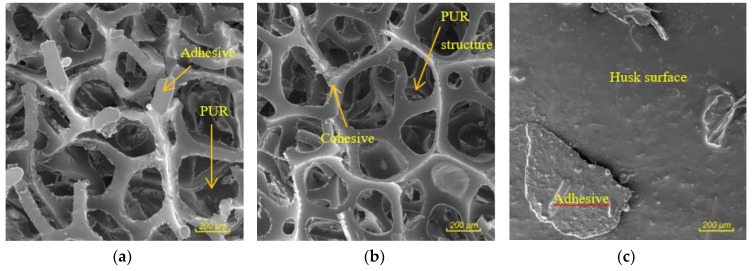
Nature of bond failure after internal bond strength test: (**a**) adhesive failure between the husk and PUR particle; (**b**) cohesion failure in PUR particle; (**c**) adhesive failure between the adhesive and husks surface.

**Table 1 materials-12-03075-t001:** Insulation board variants.

Variant	a	b	c	d	e
**Ratio of foam: husks: adhesive**	80:0:20	60:20:20	40:40:20	20:60:20	0:80:20
**Weight of PUR (g)***	320	240	160	160	0
**Weight of husks (g)***	0	80	160	480	900
**Weight of adhesive (g)***	80	80	80	160	225
**Total weight (g)**	400	400	400	800	1125
**Target thickness (mm)**	50	50	50	50	50

* Weight dry matter.

**Table 2 materials-12-03075-t002:** Actual production parameters.

Variant	a	b	c	d	e
**Weight of PUR (g)**	320.49	240.53	163.19	163.72	0
**Weight of husks (g)**	0	80.46	171.28	513.78	968.37
**Weight of adhesive (g)**	82.715	80.44	81.25	165.65	230.77
**Total weight (g)**	403.455	401.43	415.72	843.15	1199.13
**Resulting density (kg/m^3^)**	52.8	51.71	51.64	93	156.16
**Resulting thickness (mm)**	47.79	48.59	50.32	56.71	48
**Pressing pressure (kPa)**	1.25	1.25	0.3125	0.3125	0.3125

**Table 3 materials-12-03075-t003:** Parameters of insulation boards.

Characteristic	Arithmetic Mean					Standard Deviation				
**Ratio foam: husks: adhesive**	80:0:20	60:20:20	40:40:20	20:60:20	0:80:20	80:0:20	60:20:20	40:40:20	20:60:20	0:80:20
**Density (kg/m^3^)**	55.9	53.4	55.5	92.2	153.4	1.9	4.9	8.8	16.2	6.9
**Internal bond strength (kPa)**	31.1	21.5	3.4	2.3	5.8	5.9	3.0	1.4	0.8	2.3
**Compressive stress at 10 % strain (kPa)**	2.9	2.7	1.4	5.5	18.3	0.2	0.6	0.2	2.1	5.0
**Water uptake (kg/m^2^)**	4.32	4.30	3.91	3.82	3.75	0.75	0.89	0.47	0.26	0.63
**Water uptake (%)**	132.4	131.2	123.6	55.1	46.2	21.8	29.7	18.9	7.2	7.9
**Coef. of thermal conductivity (W/(m.K))**	0.0418	0.0434	0.0452	0.0497	0.0574	0.0011	0.0003	0.0009	0.0027	0.0030
**Volumetric heat capacity (J/(m^3^K))**	1.05 × 10^5^	1.08 × 10^5^	1.13 × 10^5^	1.54 × 10^5^	2.37 × 10^5^	5.19 × 10^3^	4.12 × 10^3^	1.45× 10^4^	3.22 × 10^4^	2.42× 10^4^

**Table 4 materials-12-03075-t004:** *p*-values of Tukey post-hoc test: statistical significance of the differences in the graph in Figure 2.

Ratio	80:0:20	60:20:20	40:40:20	20:60:20	0:80:20
**80:0:20**		0.226941	0.000722	0.000129	0.000129
**60:20:20**	0.226941		0.193619	0.000129	0.000129
**40:40:20**	0.000722	0.193619		0.000131	0.000129
**20:60:20**	0.000129	0.000129	0.000131		0.000129
**0:80:20**	0.000129	0.000129	0.000129	0.000129	

**Table 5 materials-12-03075-t005:** *p*-values of Tukey post-hoc test: statistical significance of the differences in the graph in Figure 3.

Ratio	80:0:20	60:20:20	40:40:20	20:60:20	0:80:20
**80:0:20**		0.998793	0.893436	0.000130	0.000129
**60:20:20**	0.998793		0.969250	0.000131	0.000129
**40:40:20**	0.893436	0.969250		0.000159	0.000129
**20:60:20**	0.000130	0.000131	0.000159		0.000129
**0:80:20**	0.000129	0.000129	0.000129	0.000129	

**Table 6 materials-12-03075-t006:** *p*-values of Tukey post-hoc test: statistical significance of the differences in the graph in Figure 4.

Ratio	80:0:20	60:20:20	40:40:20	20:60:20	0:80:20
**80:0:20**		0.000131	0.000131	0.000131	0.000131
**60:20:20**	0.000131		0.000131	0.000131	0.000131
**40:40:20**	0.000131	0.000131		0.946138	0.538191
**20:60:20**	0.000131	0.000131	0.946138		0.183404
**0:80:20**	0.000131	0.000131	0.538191	0.183404	

**Table 7 materials-12-03075-t007:** *p*-values of Tukey post-hoc test: statistical significance of the differences in the graph in Figure 5.

Ratio	80:0:20	60:20:20	40:40:20	20:60:20	0:80:20
**80:0:20**		0.999730	0.647655	0.136628	0.000134
**60:20:20**	0.999730		0.760578	0.091390	0.000134
**40:40:20**	0.647655	0.760578		0.004316	0.000134
**20:60:20**	0.136628	0.091390	0.004316		0.000134
**0:80:20**	0.000134	0.000134	0.000134	0.000134	

**Table 8 materials-12-03075-t008:** *p*-values of Tukey post-hoc test: statistical significance of the differences in the graph in Figure 6 left.

Ratio	80:0:20	60:20:20	40:40:20	20:60:20	0:80:20
**80:0:20**		0.999993	0.621564	0.421637	0.296578
**60:20:20**	0.999993		0.669087	0.467136	0.335045
**40:40:20**	0.621564	0.669087		0.997587	0.980673
**20:60:20**	0.421637	0.467136	0.997587		0.999394
**0:80:20**	0.296578	0.335045	0.980673	0.999394	

**Table 9 materials-12-03075-t009:** *p*-values of Tukey post-hoc test: statistical significance of the differences in the graph in Figure 6 right.

Ratio	80:0:20	60:20:20	40:40:20	20:60:20	0:80:20
**80:0:20**		0.999907	0.840856	0.000134	0.000134
**60:20:20**	0.999907		0.901473	0.000134	0.000134
**40:40:20**	0.840856	0.901473		0.000134	0.000134
**20:60:20**	0.000134	0.000134	0.000134		0.836946
**0:80:20**	0.000134	0.000134	0.000134	0.836946

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
