# Peer review of "Waste Utilization: Insulation Panel from Recycled Polyurethane Particles and Wheat Husks"

_materials, 2019, doi:10.3390/ma12193075_

Round 1

Reviewer 1 Report

There are a number of minor language errors through the text - eg I found some on lines 28, 50, 53, 93 and 94, but I noted more. I would recommend getting this done to improve the paper.

The paper considers the development of insulation panels for buildings from waste husk and polyurethane foam. Although not written into the paper this is a good example of a circular economy approach to buildings and is very contemporary. The experimental design with the test materials (glue, foam and husks in different ratios) is clear and provides results for thermal properties, stress and water uptake.

Author Response

Dear reviewer,

thank you very much for your valuable suggestions. Please find enclosed to this cover letter the revised manuscript and responses to your comments.

I believe that after carefully performed revisions, the manuscript deserves to be considered for publication in the journal Materials.

Best regards,

Štěpán Hýsek

Reviewer 2 Report

This manuscript presents the production of composites made of two waste materials, namely the residues of soft polyurethane foam from the production of mattresses and winter wheat husks.

This contribution is very interesting taking into account the utilization of this kind of wastes in construction materials and also the increase of the incorporation of bio-based material in insulation boards. However, it would be interesting to compare the properties with commercial materials, as polystyrene boards or wood fiberboards used for the same purpose.

The introduction gives a good overview of the topic, but in material and methods section there is information that is missing. The conditions for compression moulding are missing, as the pressing pressure. Only the target thickness is indicated. Is there any expansion of the material (spring-back) when the pressure is removed ? What material was used in the mold ? Although a commercial adhesive was used, it would be interesting to present the basic properties of the adhesive. All the boards were produced with same adhesive batch ?

The internal bond strength was measured according to EN 319 used for wood-based panels. Why the thickness swelling test was not carried out according to EN 317? It would be interesting to compare this property with low density wood-based panels. This will improve 

The compressive strength was measured according to EN 1607:2013 Thermal insulating products for building applications. Determination of tensile strength perpendicular to faces; Eur. Comm. for Stand.: Brussels, Belgium, 2013. Is it correct ?

The results are interesting for a preliminary study, since it presents only the effect of material formulation (materials weight ratio). In comparison to wood-based panels used as insulation material, the time of production and the adhesive content are much higher. Furthermore, the board final properties are very dependent on density. The density of boards produced with different proportions of foam-husks-adhesive are very different. It could be very interesting to perform an ANOVA in order to compare also the effect of the density. It could be also interesting to maintain the target density and compare the properties.

Authors should provide more information/clarify or revise the following suggestions indicated:

Line 122 please correct “internal bonding” for “Internal bond strength”

Line 141 “variance analysis” should be corrected to ANOVA (Analysis of variance).

Line 179 please explain what is the meaning of the λ20/65 in plot captions.

Line 181 please correct P-value to p-value

Line 189 Correct cp (heat capacity) to Cp (p is always in subscript and indicates constant pressure, C should be in italic).

Line 189 Correct the units MJ/m3.K to MJ/(m3K) or MJ m-3 K-1

Line 275 Correct “Tensile strength test perpendicular to the level of the board” to “Tensile strength test perpendicular to the plane of the board”

So, I recommend the acceptance of the paper with major revisions.

Author Response

(The authors gave the same response as above.)

Reviewer 3 Report

See attached pdf file. Somehow my file didn't have line numbers and so I trust that the authors can follow my attempt to be clear as to where each of my suggestions belongs

Author Response

(The authors gave the same response as above.)
